# Development and Testing of a Soft Exoskeleton Robotic Hand Training Device

**DOI:** 10.3390/s23208395

**Published:** 2023-10-11

**Authors:** Gregory Jackson, Hussein A. Abdullah

**Affiliations:** The Robotics Institute, School of Engineering, University of Guelph, Guelph, ON N1G 2W1, Canada; jacksong@uoguelph.ca

**Keywords:** rehabilitation robotics, hand rehabilitation, exoskeleton device, stroke rehabilitation, pneumatic actuator

## Abstract

Hand-function recovery is often a goal for stroke survivors undergoing therapy. This work aimed to design, build, and verify a pneumatic hand training device for its eventual use in post-stroke rehabilitation. The system was built considering prior research in the field of robotic hand rehabilitation as well as specifications and design constraints developed with physiotherapists. The system contained pneumatic airbag actuators for the fingers and thumb of the hand, a set of flex, pressure, and flow sensors, and software and hardware controls. An experiment with the system was carried out on 30 healthy individuals. The sensor readings were analyzed for repeatability and reliability. Position sensors and an approximate biomechanical model of the index finger were used to estimate joint angles during operation. A survey was also issued to the users to evaluate their comfort levels with the device. It was found that the system was safe and comfortable when moving the fingers of the hand into an extension.

## 1. Introduction

The use of robotics to facilitate physiotherapy treatment for neurological impairments has prompted the development of robotic devices for rehabilitation over the last 20 years. Sufferers of various ailments have regained motor ability in paretic limbs via highly repetitive, intense movement therapy [1,2]. Robotic systems have been shown to be able to administer such therapy accurately while producing quantitative sensor data for evaluation [3,4]. Robotic therapy has been shown to create clinically significant improvements in upper limb motor movement for those who have suffered a stroke [5,6,7].

Similar robotic devices have been developed to rehabilitate and strengthen impaired hands to assist patients in achieving daily living (ADLs) [8]. They are used under the guidance of a physical therapist to help patients regain greater functionality. They can be classified as orthoses, exoskeletons, or end-effector devices [9]. Orthoses act like a hand brace, mainly supporting the hand [9]. In contrast, exoskeletons and end-effector devices are powered devices that provide support and actuation of the hand to do passive and active therapy exercises [1,9,10]. They provide patients with rehabilitation using different design architectures, sources of actuation, mechanisms, and sensory and control systems. An intensive review and evaluation of the robotic hand rehabilitation devices and their research status were published by Kabir et al. [9] and Aggogeri et al. [10]. Moggio et al. [1] published a meta-analysis comparing exoskeleton versus end-effector robot-assisted therapy for finger and hand motor recovery in stroke survivors.

Exoskeleton devices are designed as wearable electromechanical devices [1]. Comprehensive listings and reviews of exoskeleton hand rehabilitation devices have been published by Bos et al. [11] and Heo et al. [12]. Due to the complex structure of the hand and the high number of degrees of freedom (DOFs), many exoskeleton devices have been designed with large numbers of actuators and sponsors to be able to manipulate all the fingers. As a result, exoskeleton devices became heavy, costly, complex, and uncomfortable when used [8]. Therefore, research was directed toward developing soft exoskeleton devices as alternative designs to mitigate these issues by placing the actuation and sensory systems away from the hand. In addition, soft gloves with a cable-driven mechanism or pneumatic actuation were used for hand rehabilitation to reduce the size and weight of the system [3,13,14,15,16,17]. Soft exoskeleton devices can provide a simple mechanism structure, are not rigid, are lightweight, and cost less.

Pilot studies have been reported and include a variety of approaches. Occupational therapists have found that the glove used to attach the user to the system would be difficult to get onto the hand of a spastic patient [18]. Other studies found that using gloves can have limitations due to different hand sizes and could induce significant joint reaction forces [19,20]. Many papers regarding systems for hand rehabilitation have been published, yet it appears that many of these designs have not made it to clinical trials [21].

Designers of robotic hand rehabilitation systems must work with clinicians in order to create clinically acceptable solutions that can move on to clinical testing [7]. Some of the systems presented in the literature are not without drawbacks, even though they have been developed and reached various levels of testing. Some systems in hand rehabilitation do not actuate the thumb for training [22,23]. This factor is critical for people to regain motor ability in their paretic hands. Some glove-based systems have been difficult to attach to subjects [18,24]. Following discussions with physiotherapists, the location of the actuation became an important issue. They suggested that the actuation mechanism not be located on the palmar side of the hand so that the hand may be free to interact with real-world objects during therapy. Not all prior work has left the palm of the hand free for rehabilitation exercises [25].

Upon consideration of these factors, the previous work outlined above and our earlier research experience with cable-drive hand devices [26]. The main objective of this study was to develop and test a soft, wearable, powered exoskeleton for the human hand to improve the structural stability of the fingers while enhancing pinching and grasping efforts. The user’s pinching and grasping efforts are essential to supporting the ability to perform ADLs independently. The proposed exoskeleton device is pneumatically actuated and provides a light, soft, comfortable, and user-friendly design. The system addressed clinician demands and some of the limitations of the previous designs reported in the literature. It can actuate the thumb and can be easily attached to the dorsal side of the hand. It also does not rely on a glove-based actuation and cable-driven system, which allows the system to be easily attached, be comfortable, and be adjusted to fit a variety of hand sizes.

The first step taken to create this system was to consult physiotherapists to receive input on deciding a set of constraints and criteria that should be applied to the design. The second step was to build the design while meeting all the constraints and addressing some of the perceived limitations in previous works. The third step was to conduct an experiment on healthy individuals to demonstrate that the system was safe, comfortable, and met the design specifications necessary to get approval from regulatory bodies for clinical testing with stroke subjects.

## 2. Materials and Methods

### 2.1. System Design Overview

Based on consultations with physiotherapists, the following constraints and criteria were established for the main design specifications:Constraints:
–The maximum weight of the design would not exceed 800 g. The physiotherapists selected this value as the maximum weight at which a hand device would not fatigue the user during operation.–The maximum length of the wearable device would not exceed 30 cm, a value selected by the physiotherapists so that the device would not impede the movement of the hand and arm during operation.–An emergency stop would be readily available in the event of any need to interrupt the therapy exercise.



Criteria:
–Emulating natural range of motion (ROM) as closely as possible.–Adhering to all regulations established by regulating bodies for medical devices.–Minimizing the cost of manufacturing.–Easy to attach with a minimum setup time.


After reviewing the work presented in the literature, it was found that most of the systems that made it to clinical trials used either mechanical or pneumatic actuation. Based on the system’s specifications, force is required to open the impaired hand. After consulting with physiotherapists, pneumatics was selected as the primary method to actuate the system. Pneumatics deliver a large amount of force while remaining lightweight compared to mechanical actuation systems. This would help to build a device weighing less than 800 g.

### 2.2. System Hardware Components

The five main hardware modules used in the system were as follows:Airbags: The airbags would be attached to the user’s hand and actuated to open them.Pneumatic Hardware and Sensors: These components monitor and control the airflow into the bags. They included a solenoid, airflow and pressure sensors, electro-pneumatic regulators, filters, and an emergency release valve. Data from the sensors within this module would be sent to the National Instrument Data Acquisition (DAQ) board to provide trial and safety information. Control signals regulating the airflow into the bag would be sent from the DAQ via the interfacing circuitry.Position Sensors: Two different types of position sensors were integrated into the system. The 3D pose Polhemus Patriot sensors and goniometers (flex sensors) were used in order to control the position of the fingers of the hand during operation.Interfacing Circuitry: The electric circuits of this module would control the pneumatic hardware. This circuitry also integrated the sensors and their signals with the main hardware modules of the other five system modules.Data Acquisition: This module would receive sensor data to be used by the software for system control and data gathering purposes. It would also send out control signals to the interfacing circuitry to control the pneumatic hardware.

The system configuration and how each component interacts can be seen in Figure 1. A set of pneumatic airbags was designed to be attached to the dorsal side of the user’s hand to actuate the fingers and thumb for training. The sensors on the airbag relayed information via the interfacing circuitry to the data acquisition (DAQ) system. The volume of air that inflated the airbags was regulated by pneumatic hardware and sensors, which also relayed information to the DAQ. The DAQ interfaced with a personal computer (PC) to record sensor data and control airflow into the airbag actuators.

#### 2.2.1. Pneumatic Actuators and Position Sensors

A pair of pneumatic airbags were designed to extend the fingers of the user’s hand. Pneumatic actuation was selected for numerous reasons. It has a high force-to-weight ratio, meaning a great amount of force can be exerted on the hand without weighing it down. It has also been successfully used in other robotic rehabilitation systems [19].

To determine the appropriate size of the bags, an anthropomorphic study conducted by the United States Marine Corps (USMC) was consulted [27]. This study outlined various bodily dimensions of men and women in the USMC and acted as a guideline to determine the airbags’ dimensions. The maximum values are the 95 percentile male size measurements, and the minimum values are the 5th percentile female. This data showcased the wide discrepancy in hand and wrist measurements between males and females. Accordingly, the dimensions of the developed bags were determined as follows:Finger airbag actuator: 26 cm × 12 cm;Thumb airbag actuator: 18.5 cm × 4.5 cm.

The dimensions of the finger bag would ensure that the maximum size of the hands was accounted for. The thumb bag was long enough to account for the most extended wrist thumb tip length, 15.1 cm, by being 18.0 cm long. Using the Velcro pads attached to the bags would also allow users with smaller fingers and wrist–thumb-tip lengths to use the device.

The airbag prototype design was manufactured in the lab. Both bags had a vinyl covering, a pneumatic bulkhead, and a polyurethane inner tube. The inner tube of both bags was made of a single piece of polyurethane that was cut such that when it was folded over in half, it would match the dimensions of the vinyl cover. A Uline heat sealer was used to seal the polyurethane to form a customized bag that was virtually airtight. The airbags used in the system prototype are shown in Figure 2 and Figure 3. Figure 2 presents the front and back images of the finger bag components. Figure 3 displays the front and back images of the thumb bag elements.

Figure 4 shows the palmar and dorsal views of the subject wearing the finger and thumb airbags. Two basic principles were followed in developing the actuation system. The first was that the fingers did not have to be actuated individually to provide retraining for grasping tasks. The second was that the actuation system should be located on the dorsal side of the hand so that the palm remains free from obstruction.

The physiotherapists suggested this second point, as anything on the palmar side of the hand would not allow the user to grab real objects and train for ADL. With these concepts in mind, a pair of pneumatic airbag actuators were developed: one to actuate the index, middle, ring, and little fingers simultaneously, and one to actuate the thumb. The bags were comprised of a vinyl outer shell, a polyurethane inner tube, and a lightweight pneumatic bulkhead.

The attachment of the airbags to the hand can be seen in Figure 4. The finger bag actuator was attached to the hand via straps at two locations: one at the wrist and one at the distal joints of the index, middle, ring, and little fingers. The wrist strap was woven through the vinyl cover of the actuator and then wrapped around the user’s wrist. The finger strap was wrapped around the distal joints and then secured to the vinyl cover by Velcro strips that were attached to the vinyl cover. The actuator straps for the thumb’s airbag were located at the same positions: at the wrist and distal joint of the thumb. The Velcro strips located on the vinyl bag allowed the user’s fingers to be placed anywhere along the length of the bag. This meant that the device could accommodate several different hand sizes. The Velcro strips provided a quick setup time, allowing the device to be adjusted quickly during setup, which rehabilitation personnel prefer.

The airbags were able to pull the fingers of the user’s hand into extension rather than both extension and flexion. Through discussions with the physiotherapists, it was noted that getting the user’s fingers into extension was difficult when compared to flexion, as the fingers of most stroke survivors naturally maintain a flexed position. It was decided that to keep the device simple, its operation would focus on extension movements only, as the flexion motion would be the reaction to the extension motion provided by the device.

Having the pneumatic actuators on the dorsal side of the hand prevents hyperextension of the fingers. As the airbag inflates, it becomes firmer and pulls the fingers into extension. With the fingers in this extended position, the airbag full of air acts like a wall, preventing hyperextension. Once the air is vented from the airbag, the fingers may naturally return to a flexed position.

The bags were lightweight, weighing only 180 g. An experiment was conducted on stroke patients by physiotherapists, and they determined that the maximum amount of force required to open a stroke patient’s impaired hand was 25 N [26]. This system was found to be able to lift a 6.2 lb object, exerting 27.58 N to do so. It is clear that the system can deliver enough force to actuate the hand, but to ensure that the system is safe and comfortable, the device needed to be tested on subjects for safety and comfort.

Spectra (goniometer) flex position sensors manufactured by Spectra Symbol were also attached to the bags to control the airflow based on the fingers’ positions. These 4.5″ sensors were located on the portion of the airbags that are pressed against the fingers and thumb of the hand and are attached to the finger straps via Velcro tape. The voltage from these sensors was read via the DAQ, and then, leveraging the National Instruments API, the software used this value to control if the airflow was on or off. To calibrate the sensors prior to starting the trials, an initial measurement was taken with the bag attached to the fingers in a flexed position. From the specifications of the sensors, the nominal or non-bent voltage of the sensor was known. If the DAQ was reading the nominal voltage, it would indicate that the sensor is fully extended and, therefore, the fingers are extended. Knowing this initial position and the non-bent value of the sensor, the sensor reading at the flexed and extended positions of the fingers is used to toggle the airflow into the actuators (on and off).

#### 2.2.2. Pneumatic System Hardware Components

The section presents the main components of the pneumatic hardware selected to build the developed system prototype. The selected components and everything needed to interface the components, including brackets and mounts, were sourced from SMC pneumatics. A 1/4″ National Pipe Thread (NPT) and 3/8″ push-in fittings were chosen to connect the pneumatic hardware components.

Figure 5 shows the schematic of the main components of the system’s pneumatic circuit. Particle and moisture filters were used to remove particulates and moisture from the airflow so that the air remained clean and did not damage any other equipment. Having moisture and particles in the compressed air may lead to degradation of the system’s electrical, electro-mechanical, and airbag components. These filters have a maximum operating pressure of 125 psi.

A VHS20 switch, seen in Figure 5, is connected to these filters. This valve can be switched to vent the incoming air into the atmosphere in an emergency. A 1/4″ NPT muffler is placed on the port, venting to the atmosphere to prevent particles from entering the system. Having an emergency stop is critical in the event of a software failure. An analog pneumatic regulator and a digitally controlled regulator were placed in series behind the filters and emergency stop. The analog regulator was a Topring 1/4″ R 62.125 regulator. The digital regulator was the ITV1030-21N2N4 electro-pneumatic regulator. This two-stage pneumatic regulation was selected instead of a single-stage regulator. The analog regulator reduced the high pressure at the supply, approximately 100 psi, to a usable pressure of less than 75 psi for the digital regulator. The digital regulator produced fine control and lowered the pressure more accurately to 5 psi. The digital regulator also allowed an analog input voltage to control the incoming air pressure.

Two NVS pneumatic solenoids were placed after the regulators. They were used to start and stop the regulated flow of air into the finger and thumb airbags. Each solenoid is a 3-port/2-way solenoid with a maximum working pressure of 150 psi. When it is engaged, the solenoid allows for airflow to the airbag. When it is not engaged, it vents the incoming air out of the system. A muffler was placed on the port connected to the atmosphere to prevent particles from entering the valve. The solenoid requires a 24 V signal to energize the coil and switch the port configuration.

In order to determine the flow of air into the system, a PF2A flow sensor was integrated into the system. This sensor was connected to a digital readout displaying the flow rate as well as to the DAQ to collect the output voltage. A PSE532-M5 pressure sensor was integrated with the system to measure the pressure in the pneumatic airbags. The pressure readings from the sensors were displayed on a digital display, and the DAQ read was the analog output voltage of the sensor. A National Instruments USB 6009 DAQ (Texas Instruments, Dallas, TX, USA) was used to interface a personal computer and its control software with the system’s hardware components. The personal computer acts as the system’s central controller and hosts the Graphical User Interface that users use to interact with and control the soft exoskeleton robotic hand training device.

### 2.3. Software System

The software system was designed and developed to control the pneumatic bags and record the three sensors data (flex, flow, and pressure). It was based on a modular design approach. The software was coded using a C# program developed in Visual Studio. The use of Visual Studio for development made it easy to integrate a Microsoft Access (Microsoft, Redmond, WA, USA) database for data storage and the National Instruments software libraries for interaction with the DAQ. An image of the basic software architecture can be seen in Figure 6.

The software system consisted of three levels:Interface Level: This software level contains the graphical user interface (GUI) to allow users of the soft exoskeleton device to set up and initiate trial parameters. It interacted with the controller level by sending trial parameters and subject information to be used to initiate trials. The controller level sent updates to the interface level regarding the trial status during operation. The software user could also retrieve subject information from the system. The GUI will collect the following parameters: trial ID, type of training, date and time of the trial, number of repetitions, time of completion, and notes about the trial.Controller Level: This software level controls how the trials are executed. Based on the data sent from the interface, the trials are initiated, and their status is relayed to the system users. The controller level sends commands and data requests to the handler level to complete the trials.Handler Level: This software level contains three sub-modules that interface the software with the hardware and the database, as seen in Figure 6. The database handler receives commands from the controller level to read, write, query, and delete data in the database. The sensor handler gathers sensor data and sends it to the Controller Level for control. The air handler controls the valves that allow air to flow into the system for hand actuation.

The software was organized into these modules for abstraction purposes. The GUI does not know any of the implementation details of the Handlers. This means that if the sensors, DAQ, or database are changed, it will not affect the user experience. If the GUI is modified, it will not affect the operation of the low-level Handlers.

### 2.4. Experimental Setup

The pneumatic hardware used to monitor and control the airflow into the pneumatic actuators was acquired from SMC Pneumatics, and the overall arrangement of the components can be seen in Figure 7. The components were arranged in series, with the air flowing from the regulators all the way to the airbag.

The user could load and create information regarding each subject by applying the system’s graphics user interface (GUI). This permitted the user to track a subject’s progress over time.

The graphics interface was also able to select various parameters for the trials (Figure 8). This included the number of flexion and extension repetitions and the type of training for finger, thumb, or finger and thumb training. The interface also contained an advanced interface that could diagnose any issues with the operation and settings of the system. There was also an emergency software stop button that could be selected in the event of an emergency.

#### Experimental Goals

The overall aim of the experiment (trial) that was carried out was to prove the working principles of the device and to determine that the system design was safe and comfortable. This initial testing was performed on healthy individuals who could perform both flexion and extension under their own volition. This population was selected for initial testing to gather data about the performance and comfort of the device prior to applying for approval from regulatory bodies for clinical testing. To do this, a set of experimental goals were established:i.Ensure the system does not place the fingers at non-biomechanically compatible angles. This is needed to ensure that the system does not place the finger’s joints at any angles that do not fall within the normal joint ROM.ii.Ensure that the system is comfortable for the user. A qualitative survey was carried out to evaluate how comfortable the user felt the device was.iii.Ensure the safety of the system with reliable and repeatable operation. The repeatability was measured by looking at the peak sensor values during operation to ensure that the device reached consistent sensor values throughout the experimental trials.

### 2.5. Biomechanical Model

To reach the first goal of the experiment, a biomechanical model of the hand’s index finger was created. The finger was modeled as a three-link, three-joint, serial kinematic chain. It was assumed that the finger did not experience a significant amount of abduction or adduction during the flexion and extension movements. This was carried out to simplify the model since the three-link model operated in only one plane of movement. This was based on the fact that the blade of the user’s hand would be resting on an armrest and, in doing so, would prevent any significant adduction/abduction.

An image of the model can be seen in Figure 9. Standard 4 × 4 transformation matrices were used to build the model for the three joint links and can be found in [28], page 75. The coordinate frames were placed at the joints and tip of the finger, with the base frame, X0, Y0, located at the knuckle and the final frame, X3, Y3, at the tip of the finger. Multiplying these matrices in Equations (1)–(3) together yielded the forward kinematic model that presented the relationship between the location of the base frame and the finger’s tip. This model was based on the algebraic solution presented by [28]:(1)$$T_{1}^{0}=\left[\begin{array}{cccc} \cos\theta_{1} & \sin^{-1}\theta_{1} & 0 & 0\\ \sin\theta_{1} & \cos^{-1}\theta_{1} & 0 & 0\\ 0 & 0 & 1 & 0\\ 0 & 0 & 0 & 1 \end{array}\right]$$
(2)$$T_{2}^{1}=\left[\begin{array}{cccc} \cos\theta_{2} & \sin^{-1}\theta_{2} & 0 & L_{1}\\ \sin\theta_{2} & \cos^{-1}\theta_{2} & 0 & 0\\ 0 & 0 & 1 & 0\\ 0 & 0 & 0 & 1 \end{array}\right]$$
(3)$$T_{3}^{2}=\left[\begin{array}{cccc} \cos\theta_{3} & \sin^{-1}\theta_{3} & 0 & L_{2}\\ \sin\theta_{3} & \cos^{-1}\theta_{3} & 0 & 0\\ 0 & 0 & 1 & 0\\ 0 & 0 & 0 & 1 \end{array}\right]$$

It was created to verify the angles produced by the inverse kinematic model of the system. This inverse kinematic model of the finger was created so that, by knowing the fingertip location and the intermediate joint lengths of the finger, the joint angles could be estimated. The MCP joint angle was iteratively selected, and the location of the PIP joint in space was estimated. The reference point was then moved from the base to this location in space. The two subsequent joint angles, PIP and DIP, were calculated using the two-link manipulator inverse kinematic solution in [28]. These angles were entered into the forward kinematic model. Any finger-tip points that were calculated using the inverse kinematic model that did not fall within a working tolerance of 2 cm in both the x and y axes of the originally measured point were discounted from the analysis.

Before starting the trials, each subject had the intermediate phalanges of their index finger measured. The fingertip location and the knuckle, or base, of the finger were measured using 3D Polhemus Patriot pose sensors worn during the verification and trial. The location of the fingertip was measured relative to the finger’s base by subtracting one from the other. These tip positions were entered into the inverse kinematic model, along with the joint lengths, to estimate the finger joint angles.

### 2.6. Experimental Procedures

The following procedure was completed with each subject:The experiment’s objectives, procedure, and methodology were explained to each participant.Each subject provided consent to be part of the experiment. They also provided their respective information for the analysis.The position sensors were attached to the tip and base of the participant’s index finger. The finger bag actuator was then attached.With their arm resting on an armrest, the subject then placed their fingers in a flexed position.Once the subject was ready, the system was activated, and the finger and thumb bag actuators began to fill with air, pulling the fingers and thumb into extension.Once the fingers and thumbs reached the extension, air flow into the bag was stopped.This extension/flexion movement was repeated for a total of eight repetitions.The subject placed their arm back on the armrest and placed their thumb in a flexed position.At the end of the trial, each subject was issued a comfort survey to complete.

The sensor data gathered for each subject during this trial formed the basis of the analysis of the device to meet the experimental goals. The sensor data was collected and stored in a database developed for the project using Microsoft Access. A total of 30 healthy subjects, ages 21–65, were recruited for this study. There was a 16/14 split of males and females. No participant had any health concerns that may have affected their ability to take part in the study. The experimental methodology was approved by the University of Guelph Research Ethics Board prior to the start of the study (protocol no. REB #12DC007).

## 3. Results and Discussion

In order to verify this biomechanical model, five subjects placed their index fingers in two different positions. The first position was with the finger fully extended, as shown in Figure 10. The second position was with the finger curled in towards the palm of the hand, which can be seen in Figure 11. Prior to placing their fingers in these positions, the PIP, DIP, and MIP joint lengths for each subject’s index finger were measured with the pose sensor. One sensor was placed on the tip of the finger, and another one was located at the base of the knuckle. The coordinates of both sensors were measured. These values were then entered into a MATLAB program that ran the inverse kinematic model. The error between the model-calculated (predicted) angles and the measured real angles was calculated. The computed angles and finger joint lengths were then entered into an approximate forward kinematic model to ensure that the original fingertip point could be calculated.

The results of this test can be seen in Table 1. The average absolute error for the joint angle prediction was 2.67°.

The predicted angles were then entered into the forward kinematic model to ensure that it could predict the original position of the fingertip. The predictions and errors can be seen in Table 2. The largest absolute error was 0.878 cm. This model was applied to the subject data gathered during the experiment and could estimate joint angles within a normal ROM for healthy individuals.

### 3.1. The Reliability of the Comfort Evaluation

A comfort survey was issued to the subjects after using the device to ensure that the system was comfortable. This survey had the users rank their comfort with the device on a scale from 1 to 10, with 1 being “Not Comfortable” and 10 being “Very Comfortable”. Once the surveys were gathered, they were analyzed, and the effects of age, gender, weight, and hand length were determined.

The responses to the comfort survey were positive. On average, the 30 users scored comfort at 9.04 and 8.83 out of 10 for the comfort of the finger and thumb bags, respectively. This indicates that many of the subjects found the system to be comfortable. The data recorded followed a normal distribution. A further analysis, via Student’s *t*-tests and non-parametric tests, was completed on the data to determine if there were statistically significant differences in the average scores for different groups of subjects. There was no statistically significant difference between the average comfort scores for the finger and thumb bags when accounting for age, gender, weight, and hand length (all *p* > 0.05).

Table 3 presents the t-test conducted on the comfort rankings for the finger airbag according to gender. In this example, both the parametric (Pooled) and non-parametric (Satterthwaite) tests yielded a p-value that is greater than the statistically significant threshold of *p* < 0.05 (*p* > 0.31). Table 4 presents the Kruskal-Wallis test results on the thumb airbag comfort data. The chi-square value was greater than 0.05.

The comfort survey also asked the users if the device fit their hands, and all of them indicated that the airbags could do so. The survey results indicated that the airbags were comfortable during usage regardless of the subject’s hand length, age, weight, or gender. It also suggested that the device fit the hands of all individuals who took part in the study. There was no statistically significant difference between males and females in terms of comfort with the finger device.

### 3.2. Repeatability Analysis

The repeatability and reliability of the device’s sensors, including flex, air pressure, and airflow, were analyzed to ensure that the sensor’s read voltages were the same for each of the eight repetitions of the trials. These tests were critical in ensuring that the device was safe and stable enough to be used in further trials. Three main sensor voltages were examined: peak flex sensor voltage, peak flow sensor voltage, and peak pressure sensor voltage. Sample data gathered from the flex, flow, and pressure sensors can be seen in Figure 12, Figure 13 and Figure 14, respectively. The system’s repeatability and reliability were measured based on the statistical analysis of the peak sensor readings during operation. Peak values were chosen as they would represent the values of each sensor when reaching its maximum output per cycle (repetition). The peak was selected as it had the largest amplitude present in the signal.

A total of 240 sensor readings were gathered for each of the given sensor readings. These 240 readings were broken into eight groups, one for each repetition. An ANOVA was performed comparing the peak sensor values for each of the eight repetitions of the trial. The reliability of the sensors was measured by calculating the ANOVA for each subject and determining how much of the variation in the sensor readings was attributed to differences between users.

Table 5 and Table 6 present the ANOVA between the repetitions of the finger and thumb device trials in terms of the peak sensor voltages of the flex sensor. From this analysis, there is no statistically significant difference between the peak voltage values of each repetition of the flex sensor during operation. Given the nature of the sensor, this means that the fingers and thumb were able to reach the extended position consistently during both the finger and thumb trials. The peak values for the flex sensor did not differ by a statistically significant margin for either the finger bag (*p* > 0.9887) or the thumb bag (*p* ≈ 1).

The pressure sensor peak values were tested similarly, and it was found that these values did not differ for the finger (*p* > 0.999) and thumb bags (*p* ≈ 1) as well.

The peak flow sensor readings indicated a statistically significant difference between the repetitions (*p* < 0.0001). Upon further inspection of the data’s distribution, shown in Figure 13, it was found that the average reading of the first repetition of exercises was higher than in subsequent repetitions. The axis labeled “sens” in Figure 15 is the value read from the sensor, and the one labeled “rep” is the repetition of the exercise.

It is thought that initially, before the bag is inflated and the fingers are placed into an extension, there is not a significant quantity of air in the airbags. After the fingers were placed into the extension and the airflow into the bags was stopped, residual air remained in the bag while the fingers returned to the flexed position. It is thought that this residual air would offer some resistance to the incoming air flow for subsequent repetitions.

Removing the first repetition from the ANOVA analysis and looking at repetitions 2 through 8, it was observed that there was still a slight difference in the peak value, although it was not statistically significant for the flow sensor (*p* > 0.2231). The peak flow rates during the operation of the thumb bag did not differ by a statistically significant margin (*p* ≈ 1).

The reliability of the sensors was also measured, and by doing this, the amount of variation in these readings that could be accounted for by differences in the users using the device was found. The total percentage of variation between the sensor readings accounted for by differences between system users was found for the finger and thumb bags, respectively. The reliability percentages can be seen in Table 7, with the lowest reliability being 88%, indicating a high degree of reliability within the sensor readings per subject.

## 4. Conclusions

A system designed to be used by therapists for paretic hand rehabilitation through ADL training is presented in this paper. The system was developed in consultation with physiotherapists. It is based on pneumatic airbag actuators placed on the dorsal side of the hand to extend the fingers and thumb for hand and grasp training. This lightweight, adjustable, soft, and easy-to-wear solution, which contains multiple safety features, has been tested on 30 healthy subjects to verify its working principles and emphasize the device’s comfort, reliability, and safety before clinical testing. An approximate finger model estimated that the joint angles of the index finger fell within the normal ROM of healthy individuals. Users of the system, regardless of age, gender, weight, or hand length, found the device to be comfortable. The device was also found to fit the hands of all 30 subjects satisfactorily. A statistical analysis of the flex, pressure, and flow sensor data gathered during the trial indicates a high level of repeatability in the motions of the device and opens the prospect of further development. In future work, a comprehensive clinical study should be conducted with stroke patients to confirm the highly desirable effect of incorporating this system into a rehab program and determine its unique long-term possibilities.

## Figures and Tables

**Figure 1 sensors-23-08395-f001:**
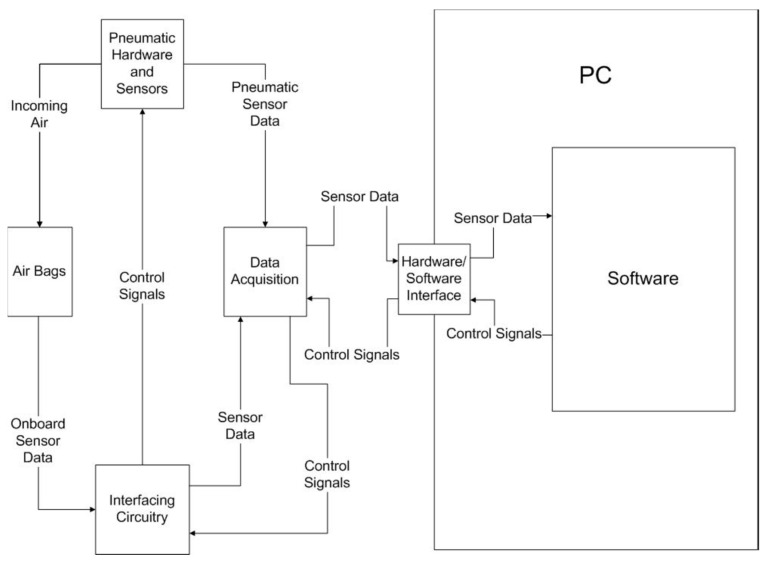
Overall view of the system.

**Figure 2 sensors-23-08395-f002:**
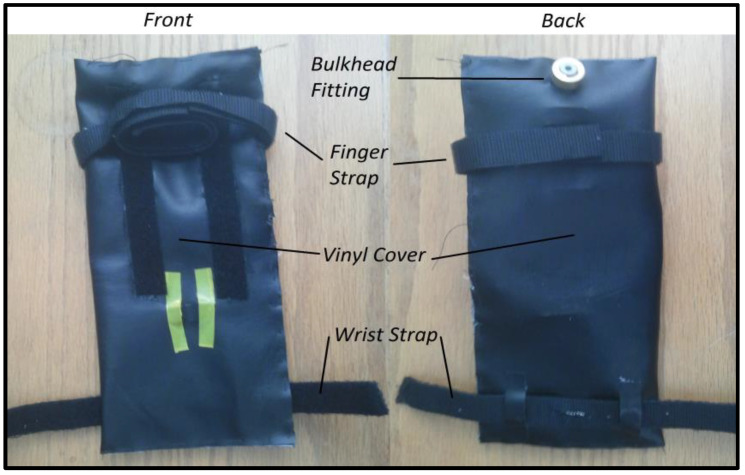
Front and back images of the finger bag.

**Figure 3 sensors-23-08395-f003:**
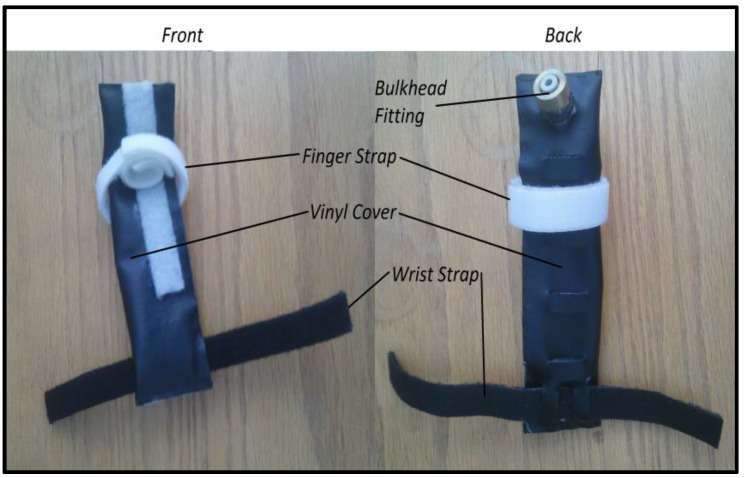
Front and back images of the thumb bag.

**Figure 4 sensors-23-08395-f004:**
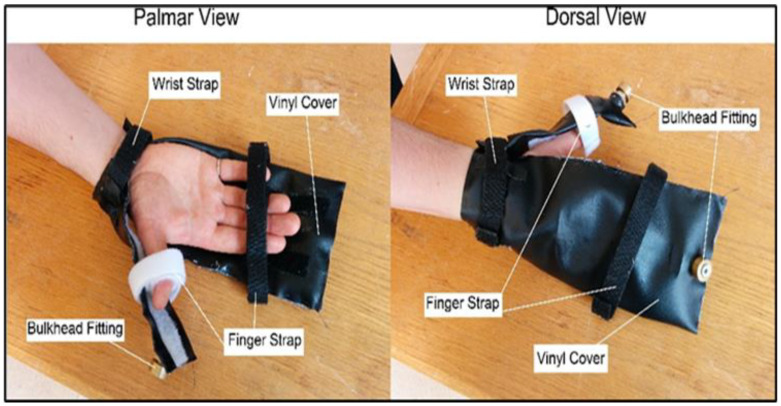
The finger and thumb airbags are used for hand actuation. Two views of the device are presented to show device attachment: palmar and dorsal. The straps, bulkheads, and vinyl covers are indicated by their respective labels.

**Figure 5 sensors-23-08395-f005:**
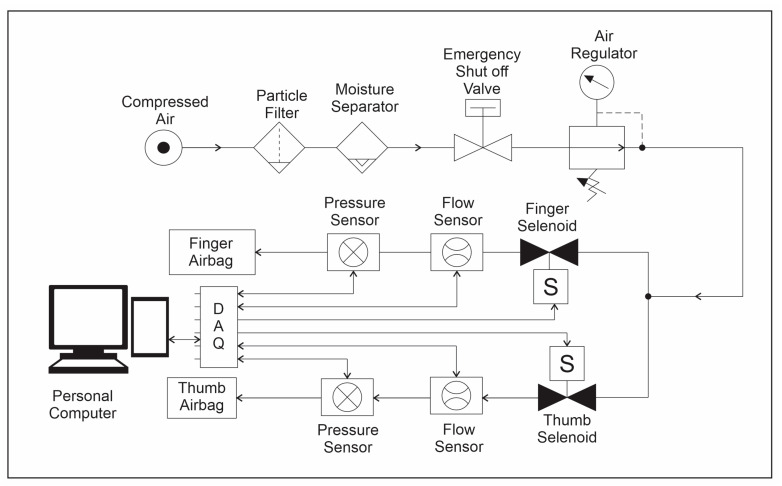
Schematic of the main components of the system’s pneumatic circuit.

**Figure 6 sensors-23-08395-f006:**
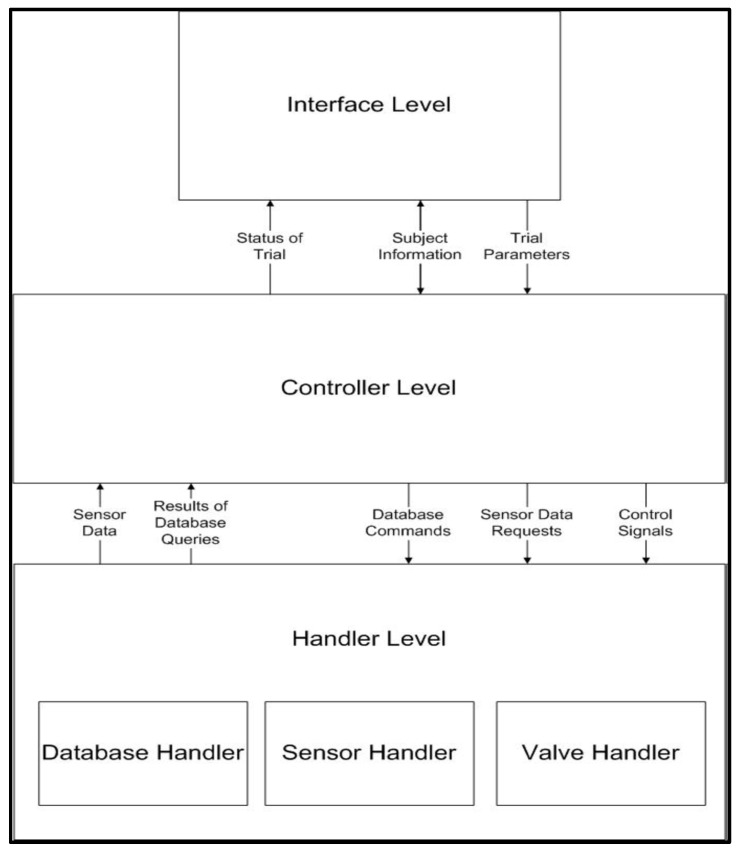
Basic software scheme.

**Figure 7 sensors-23-08395-f007:**
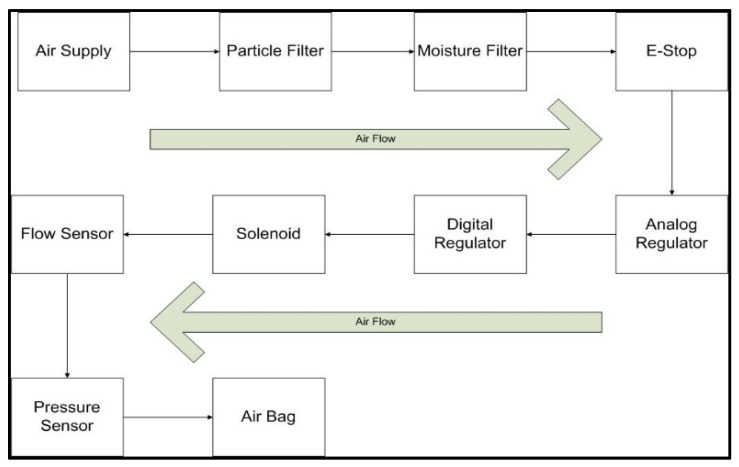
Experimental setup of a pneumatic system for airbag control.

**Figure 8 sensors-23-08395-f008:**
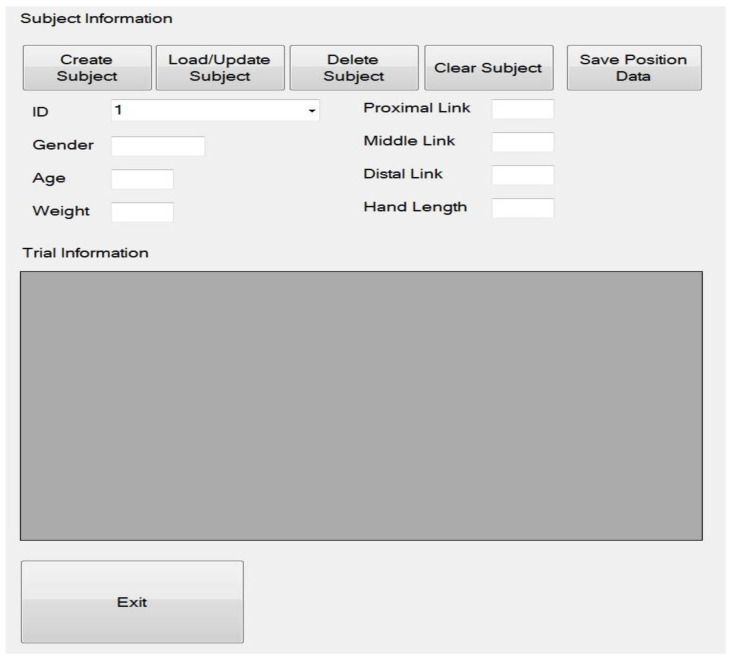
Subject information portion of the GUI.

**Figure 9 sensors-23-08395-f009:**
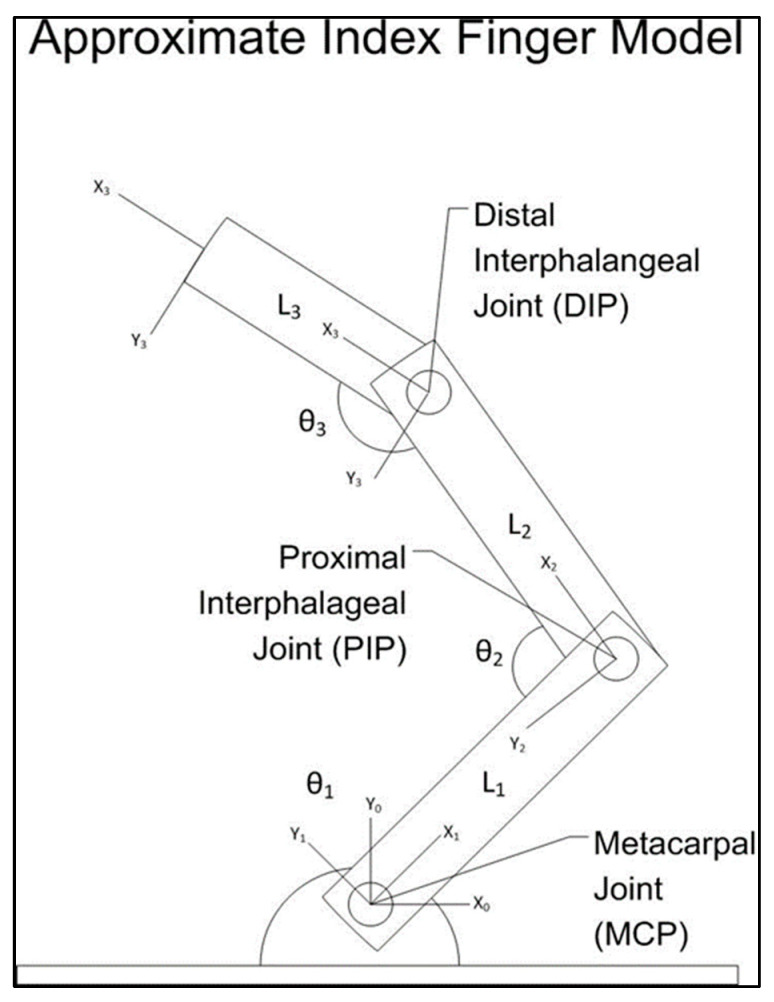
An approximate model of the index finger.

**Figure 10 sensors-23-08395-f010:**
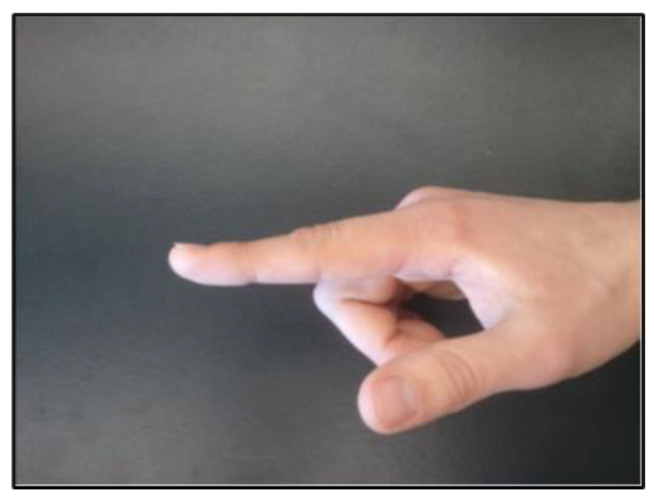
Position 1 for the Finger Model Analysis.

**Figure 11 sensors-23-08395-f011:**
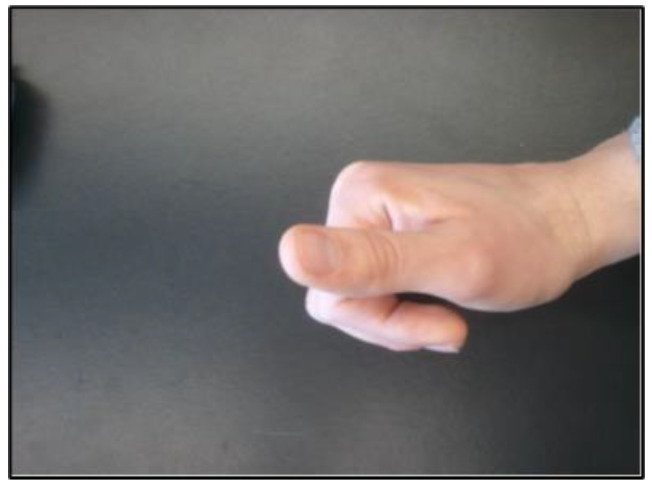
Position 2 for the Finger Model Analysis.

**Figure 12 sensors-23-08395-f012:**
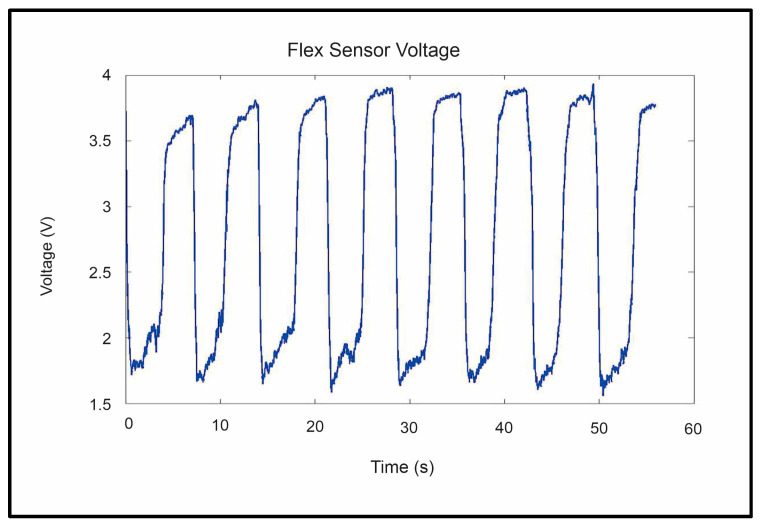
Flex sensor readings were gathered during the trial.

**Figure 13 sensors-23-08395-f013:**
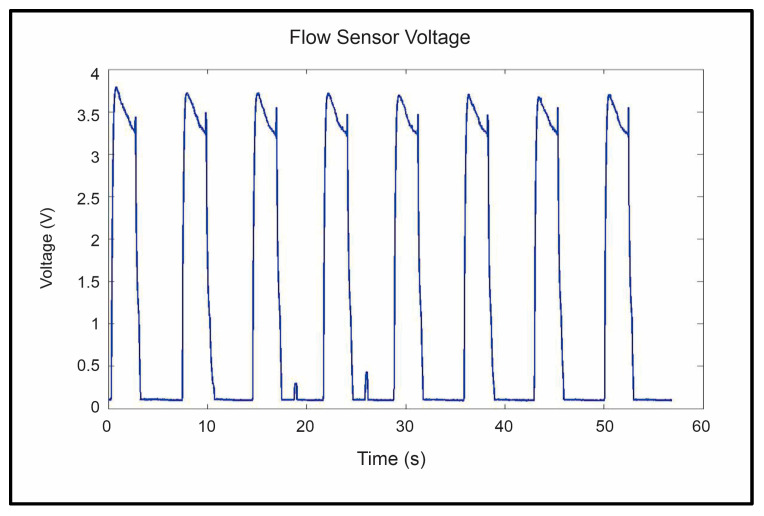
Flow sensor readings were gathered during the trial.

**Figure 14 sensors-23-08395-f014:**
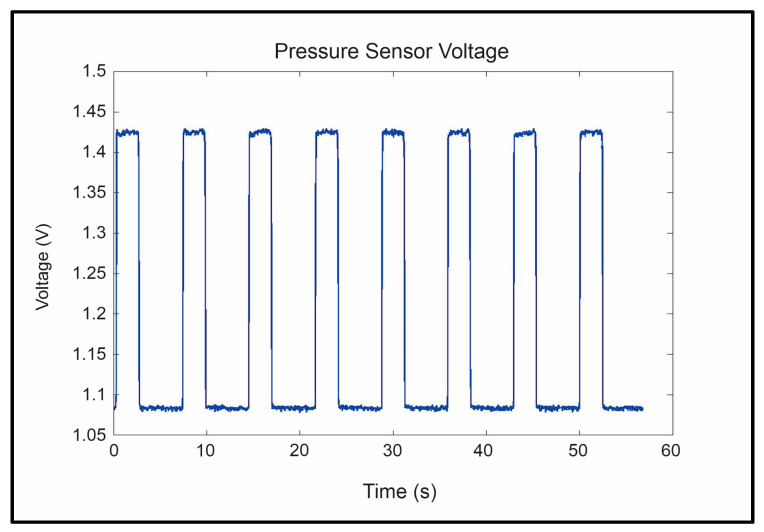
Pressure sensor readings were gathered during the trial.

**Figure 15 sensors-23-08395-f015:**
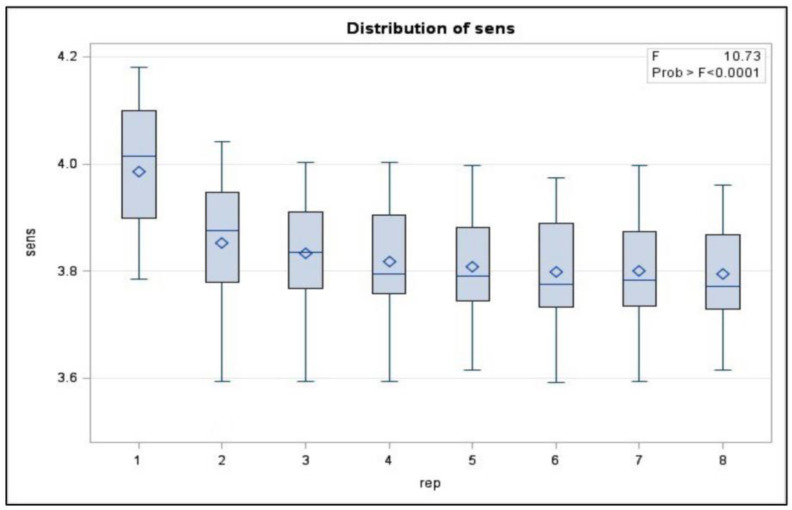
Average flow sensor readings from the experiment.

**Table 1 sensors-23-08395-t001:** Angle estimation from the approximate inverse kinematic predictive model.

Subject	Measured Angles (MCP, PIP, DIP) (°)	Predicted Angles (MCP, PIP, DIP) (°)	Absolute Error (MCP, PIP, DIP) (°)
1	90, 0, 0	87.5, 2.77, 0	2.5, 2.77, 0
	170, 80, 0	177.0, 81.5, 0	7.0, 1.5, 0
2	90, 0, 0	91.5, 9.83, 0	1.5, 9.83, 0
	170, 80, 0	167.5, 77.23, 0	2.5, 2.77, 0
3	90, 0, 0	90.5, 6.99, 0	0.5, 6.99, 0
	170, 80, 0	168.0, 77.46, 0	2.0, 2.54, 0
4	90, 0, 0	88.0, 3.43, 0	2.0, 3.43, 0
	170, 80, 0	169.0, 78.42, 0	1.0, 1.58, 0
5	90, 0, 0	93, 14.57, 0	3, 14.57, 0
	170, 80, 0	176.0, 86.3, 0	6.0, 6.32, 0
	Average Absolute Error	Position 1: 1.9, 7.5, 0	Position 2: 3.7, 2.9, 0

**Table 2 sensors-23-08395-t002:** Position estimation from the approximate forward kinematic predictive model.

Subject	Measured Position (X, Y) (cm)	Predicted Position (X, Y) (cm)	Absolute Error (X, Y) (cm)
1	0.164, 11.924	0.168, 11.158	0.004, 0.766
	−5.952, −6.589	−5.879, −6.220	0.073, 0.369
2	−1.196, 9.785	−1.345, 10.298	0.149, 0.513
	−6.559, −4.574	−6.710, −4.806	0.151, 0.232
3	−1.015, 11.552	−0.847, 10.283	0.168, 1.269
	−7.019, −5.795	−6.610, −4.801	0.409, 0.994
4	−0.136, 12.952	0.089, 11.799	0.047, 1.153
	−7.371, −4.765	−7.592, −5.200	0.221, 0.435
5	−1.984, 10.276	−2.192, 10.966	0.208, 0.690
	−5.89, −7.681	−5.672, −6.070	0.218, 1.611
	Average Absolute Error	Position 1: 0.115, 0.878	Position 2: 0.215, 0.728

**Table 3 sensors-23-08395-t003:** A T-test for measuring the differences between genders on the comfort score of the finger airbag.

Gender	N	Mean	Std Dev	Std Err	Min	Max
0	16	8.875	1.2042	0.301	6	10
1	14	9.25	0.8026	0.2145	7.5	10
Diff. (1–2)		−0.375	1.0373	0.3796		
**Method**	**Variances**	**DF**	**t Value**	**Pr *> |t|***		
Pooled	Equal	28	−0.99	0.3317		
Satterthwaite	Unequal	26.282	−1.01	0.3196		
**Equality of Variances**						
**Method**	**Num DF**	**Den DF**	**F Value**	**Pr *>* F**		
Folded F	15	13	2.25	0.1494		

**Table 4 sensors-23-08395-t004:** Kruskal–Wallis test for Thumb Airbag Comfort.

Kruskal–Wallis Test	
**Chi-Square**	2.4715
**DF**	1
**Pr ≥ Chi-Square**	0.1159

**Table 5 sensors-23-08395-t005:** ANOVA for peak flex sensor voltages (V) during the finger bag trials.

Source	DF	Sum of Squares	Mean Square	F Value	Pr ≥ F
Model	7	0.12906	0.01844	0.18	0.9887
Error	232	23.3871	0.10081		
Corrected Total	239	23.5161			

**Table 6 sensors-23-08395-t006:** ANOVA for peak flex sensor voltages (V) during the thumb bag trials.

Source	DF	Sum of Squares	Mean Square	F Value	Pr ≥ F
Model	7	0.00375	0.00054	0	1
Error	232	25.6016	0.11035		
Corrected Total	239	25.6054			

**Table 7 sensors-23-08395-t007:** Reliability of sensor readings.

Sensor	Finger Bag	Thumb Bag
Flex Sensor	88.00%	99.40%
Flow Sensor	89.50%	97.80%
Pressure Sensor	97.80%	98.90%

## Data Availability

The data supporting this study’s findings are available on request.

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
