# Peer review of "Development and Testing of a Soft Exoskeleton Robotic Hand Training Device"

_sensors, 2023, doi:10.3390/s23208395_

Round 1
Reviewer 1 Report
My main concern is that the paper is on/for stroke rehabilitation but none of the testing done with stroke patients. It may be difficult to depict the actual scenario with stroke patients. It may be difficult to wear the device if hand is spastic. Difficult to generalise the results for stroke patients, suggest changing the title not to include stroke patients.
English language is generally good but there are some typos like no coordinate X4, Y4 on Fig 4. Fig 1 could be better represented by a schematic diagram. A figure for testing with device worn would be helpful. A whole paragraph repeated on pg 9 and 10. Table 4 has formatting error. Pg 14 2 through 8 not two through eight.
Reviewer 2 Report
This work aims to develop and validate a pneumatic hand training device for post-stroke rehabilitation, incorporating sensors and controls. The system, tested on 30 healthy individuals, demonstrated safety and comfort while facilitating finger movement into extension. The research is interesting; however, the manuscript requires improvement in figures and readability to enhance its overall quality. Here are my comments: All figures need to be in a higher dpi resolution for better clarity. The introduction directly dives into the topic of robotics for rehabilitation, which affects the readability. It is recommended to provide a general background to introduce other types of robots first and then narrow down the focus to your topic. For example, you can mention research on Soft robotics with deep learning, such as Soft computing-based predictive modeling of flexible electrohydrodynamic pumps, Modeling fabric-type actuator using point clouds by deep learning and bio-inspired robots like Fluidic rolling robot using voltage-driven oscillating liquid.In line 269, the reference format is incorrect. The method part should mention the specific matrix (e.g., Jacobian matrix) instead of using a vague term like “transformation matrices.”In lines 268 and 271, “X” cannot be used to represent multiplication in “4X4.” Use the appropriate multiplication symbol, and maintain the same format as Figure 1 for variables like X0, Y0, X4, Y4.The author should declare whether the data follows a normal distribution before using the T-test for analysis. If it does not, appropriate non-parametric tests like the Wilcoxon signed-rank test (for paired data) or the Mann-Whitney U test (for independent data) should be used.Distinguish the data, signal, and air flow in Figure 1 to improve clarity.Improve the font size scale in Figures 7-9 for better readability.By addressing these points, the manuscript can be significantly enhanced in quality and presentation.
Reviewer 3 Report
The paper presents a wearable, pneumatic powered exoskeleton for the human hand designed to improve the structural stability of the fingers while enhancing pinching and grasping efforts.
The paper layout is generally correct and clear, but, overall, the novelty of this work is quite limited. The technical content is limited, emphasis is placed merely on evaluating the reliability of the comfort of the studied device and on observing the the repeatability and reliability of the device’s sensors.
The authors may consider the following comments for revising the paper.
- Distribute your text evenly between the margins (Justify).
- Figure 2 should be placed closer to the text where it is first referred to.
- Figure 2 shows only images of the utilized airbags. A constructive diagram of these would be useful.
- Line 182: what is the basis of the assertion “Once the air is vented from the airbag, the fingers may naturally return to a flexed position”? A method for ensuring with certainty the flexion is carried out should be proposed.
- Figure 3: a pneumatic diagram would have been preferable.
- Figure 4 is not followed by the corresponding mathematical apparatus.
- Table 1 should be placed closer to the text where it is first referred to.
- Figures 7, 8 and 9: the font used for the values written on the axes should be increased.
- The device’s compliance should be also studied.
Round 2
Reviewer 1 Report
Happy with the revision.
Author Response
Reviewer 1 was satisfied with round 1 of the revision and the revised manuscript.
Reviewer 2 Report
The authors graciously addressed the questions we posed.
Author Response
Reviewer 2 was satisfied with round 1 of the revision and the revised manuscript.
Reviewer 3 Report
The reviewer feels that the quality of the manuscript has not been improved as much as claimed by the authors. Although some progress was made as to the editing of the article, certain concerns remained unanswered:
- *In my previous review I suggested introducing a pneumatic diagram based on symbols according to ISO 1219-1 standard, that describes the functioning of the device. I maintain this request in order to improve the clarity of the proposed system.
- * Figure 6 is not followed by the corresponding mathematical apparatus.
- *The device’s compliance should be also studied. Compliance, namely the flexibility of physical structures in response to an external force manifests if the dependency between the force developed by the pneumatic device and the displacement caused by this is of non-linear type. Typical advantages of compliance are the possibility of rapidly and efficiently storing and releasing energy or reacting instantaneously to sudden impacts, with positive effects on stability and thereby safety.
